# hENT1 Predicts Benefit from Gemcitabine in Pancreatic Cancer but Only with Low CDA mRNA

**DOI:** 10.3390/cancers13225758

**Published:** 2021-11-17

**Authors:** Karen Aughton, Nils O. Elander, Anthony Evans, Richard Jackson, Fiona Campbell, Eithne Costello, Christopher M. Halloran, John R. Mackey, Andrew G. Scarfe, Juan W. Valle, Ross Carter, David Cunningham, Niall C. Tebbutt, David Goldstein, Jennifer Shannon, Bengt Glimelius, Thilo Hackert, Richard M. Charnley, Alan Anthoney, Markus M. Lerch, Julia Mayerle, Daniel H. Palmer, Markus W. Büchler, Paula Ghaneh, John P. Neoptolemos, William Greenhalf

**Affiliations:** 1Liverpool Experimental Cancer Medicine Centre, 2nd Floor Sherrington Building, Ashton St, University of Liverpool, Liverpool L69 3GE, UK; k.aughton@liverpool.ac.uk (K.A.); nils.elander@liu.se (N.O.E.); aevans@liverpool.ac.uk (A.E.); richj23@liverpool.ac.uk (R.J.); fcampbell@doctors.org.uk (F.C.); ecostell@liverpool.ac.uk (E.C.); halloran@liverpool.ac.uk (C.M.H.); palmerd@liverpool.ac.uk (D.H.P.); paula@liverpool.ac.uk (P.G.); 2Department of Oncology, Linköping University, SE-581 83 Linköping, Sweden; 3Cross Cancer Institute, University of Alberta, Edmonton, AB T6G 1Z2, Canada; john.mackey@albertahealthservices.ca (J.R.M.); andrew.scarfe@albertahealthservices.ca (A.G.S.); 4The Christie NHS Foundation Trust, University of Manchester, Manchester M20 4BX, UK; juan.valle@christie.nhs.uk; 5Glasgow Royal Infirmary, Glasgow G4 0SF, UK; RossCarterno1@gmail.com; 6Royal Marsden National Health Service (NHS) Foundation Trust, London SW3 6JJ, UK; david.cunningham@rmh.nhs.uk; 7Austin Health, Melbourne, VIC 3084, Australia; niall.tebbutt@onjcri.org.au; 8Prince of Wales Hospital and Clinical School, University of New South Wales, Sydney, NSW 2052, Australia; David.Goldstein@health.nsw.gov.au; 9Nepean Cancer Centre, University of Sydney, Sydney, NSW 2747, Australia; Jenny.Shannon@health.nsw.gov.au; 10Department of Immunology, Genetics and Pathology, Uppsala University, SE-751 05 Uppsala, Sweden; bengt.glimelius@igp.uu.se; 11Department of Surgery, University of Heidelberg, 69047 Heidelberg, Germany; Thilo.Hackert@med.uni-heidelberg.de (T.H.); markus.buechler@med.uni-heidelberg.de (M.W.B.); John.Neoptolemos@med.uni-heidelberg.de (J.P.N.); 12Freeman Hospital, Newcastle upon Tyne NE7 7DN, UK; richard.charnley@nhs.net; 13St James’s University Hospital, Leeds LS9 7TF, UK; Alan.Anthoney@nhs.net; 14Department of Medicine A, University Medicine Greifswald, 17489 Greifswald, Germany; Markus.Lerch@med.uni-greifswald.de (M.M.L.); Julia.mayerle@med.uni-muenchen.de (J.M.); 15Medizinische Klinik und Poliklinik II, Klinikum der LMU München-Grosshadern, 81377 München, Germany

**Keywords:** 5-fluorouracil, gemcitabine, pyrimidine, biomarker, predictive marker, prognostic marker, chemotherapy

## Abstract

**Simple Summary:**

Recent clinical trials suggest that combination therapies that include either gemcitabine or 5-fluorouracil (5-FU) both give significant survival benefits for pancreatic cancer patients. The tumor level of the nucleoside transporter hENT1 is prognostic in patients treated with adjuvant gemcitabine but not adjuvant 5-FU. This work shows for the first time that hENT1 is only predictive of benefit from gemcitabine over 5-FU in patients with low levels of CDA transcript. A choice between adjuvant 5-FU based combination therapies (such as FOLFIRINOX) and gemcitabine-based therapy (e.g., GemCap) could be made based on a combination of hENT1 protein and CDA mRNA measured in a resected tumor.

**Abstract:**

Gemcitabine or 5-fluorouracil (5-FU) based treatments can be selected for pancreatic cancer. Equilibrative nucleoside transporter 1 (hENT1) predicts adjuvant gemcitabine treatment benefit over 5-FU. Cytidine deaminase (CDA), inside or outside of the cancer cell, will deaminate gemcitabine, altering transporter affinity. ESPAC-3(v2) was a pancreatic cancer trial comparing adjuvant gemcitabine and 5-FU. Tissue microarray sections underwent in situ hybridization and immunohistochemistry. Analysis of both CDA and hENT1 was possible with 277 patients. The transcript did not correlate with protein levels for either marker. High hENT1 protein was prognostic with gemcitabine; median overall survival was 26.0 v 16.8 months (*p =* 0.006). Low CDA transcript was prognostic regardless of arm; 24.8 v 21.2 months with gemcitabine (*p =* 0.02) and 26.4 v 14.6 months with 5-FU (*p =* 0.02). Patients with low hENT1 protein did better with 5-FU, but only if the CDA transcript was low (median survival of 5-FU v gemcitabine; 29.3 v 18.3 months, compared with 14.2 v 14.6 with high CDA). CDA mRNA is an independent prognostic biomarker. When added to hENT1 protein status, it may also provide treatment-specific predictive information and, within the frame of a personalized treatment strategy, guide to either gemcitabine or 5FU for the individual patient.

## 1. Introduction

Pancreatic Ductal Adenocarcinoma (PDAC) is predicted to overtake breast cancer as the second leading cause of cancer death in the USA shortly, with limited survival despite improved therapeutic options [1,2,3].

Studies by the European Study Group for Pancreatic Cancer (ESPAC) and others show that adjuvant chemotherapy with either 5-fluorouracil (5-FU)/Folinic Acid (FA) or gemcitabine following surgery improves survival [4,5,6,7,8,9,10]. Combining gemcitabine or 5-FU with other chemotherapeutics further increases survival, including gemcitabine with capecitabine [11] or 5-FU with FA, irinotecan, and oxaliplatin (FOLFIRINOX) [12]. It is evident from trials in different population cohorts that selecting therapies based on individual profiling leads to improved survival rates [13]. Genetic variations in the patient and in their tumor [14] cause different protein patterns that can stratify patients into different sub-groups [15,16,17,18,19]. Broad classification allows association with prognosis [20], but the further subdivision is needed for treatment-specific predictions.

The activity of pyrimidine-based drugs is dependent on proteins involved in the trans-membrane uptake and metabolism of endogenous and exogenous pyrimidines [21,22]. Gemcitabine is a nucleoside analog of deoxycytidine that is transported into the cell by membrane transporter proteins, a major mediator being human equilibrative nucleoside transporter 1 (hENT1). We have previously reported that high protein expression of hENT1 was associated with improved overall survival in patients treated with gemcitabine in the ESPAC-3(v2) trial population, but not in those treated with 5-FU [23].

hENT1 has less affinity for cytidine than for its deaminated form (uridine) [24], it has much less affinity for nucleobases (e.g., 5-FU) than nucleosides (e.g., gemcitabine), although it has a greater affinity for nucleobases than other nucleoside transporters [25]. Deamination of gemcitabine by cytidine deaminase (CDA) outside of the cell would increase its transport into the cell, where it can be converted back into gemcitabine or exert a direct toxic effect [26]. Deamination inside the cell increases transport out. CDA is predominantly in the cytoplasm of cells but is also seen within the nucleus [27]. CDA can also be secreted into the extracellular space [28], and although intracellular CDA is the main determinant of gemcitabine sensitivity in cell lines, even with just pancreatic cancer cell lines, secreted CDA still accounts for a substantial amount of gemcitabine metabolism [29]. In vivo, CDA is produced by cancer and stromal cells. Bacteria found in PDAC (e.g., gammaproteobacteria) also produce CDA, perhaps contributing to resistance to gemcitabine [30].

Intracellular gemcitabine is phosphorylated by deoxycitidine kinase and nucleotide kinases to its active metabolites [31]. The phosphorylated forms of gemcitabine (as with all nucleotides) are not transported by hENT1, trapping them inside the cell. They can still be deaminated by CDA, reducing cytotoxicity as fluorouridine triphosphate is less readily incorporated into DNA [32].

It is estimated that approximately 90% of intracellular gemcitabine is metabolized by endogenous CDA [33], leaving little gemcitabine triphosphate to incorporate into DNA. Germline polymorphisms of CDA have been associated with response to gemcitabine [34] as has CDA expressed from bacteria [30] and induction of CDA expression by macrophages [35,36].

In this study, the expression of CDA mRNA and protein was analyzed in tissue from patients in the ESPAC-3(v2) trial. ESPAC-3(v2) compared gemcitabine with 5-fluorouracil plus FA (leucovorin) as adjuvant therapy. Since CDA will alter the import and export of gemcitabine and its metabolites into cells by hENT1, and hENT1 expression is known to be predictive for gemcitabine efficacy, we further assessed the combined predictive value of CDA with hENT1 expression.

## 2. Materials and Methods

### 2.1. Study Design

Translational analysis of ESPAC-3(v2) was granted ethical approval by the Liverpool Research Ethics Committee (07/H1005/87). Good Clinical Practice Standard Operating Procedures were employed throughout. The trial was originally analyzed on an intention-to-treat basis but for the translational study, patients in the treatment arms were included only if treatment was received [4,5,10]. This study was conducted in accordance with REMARK criteria [37].

### 2.2. Tissue Microarray Manufacture

Tissue microarrays (TMA) were manufactured as previously reported [23]. Arrays contained cores from 434 patients, 88 patients in duplicate per array, and a total of 4–8 cores per patient across arrays. Tumor regions were identified by an experienced pancreatic pathologist (FC) using haematoxylin and eosin-stained sections. Each core on each TMA was coded and linked separately to trial identifiers ensuring blinding of the analysts to outcome and treatment.

### 2.3. RNAscope^®^ In Situ Hybridization (ISH)

Four µm TMA sections were baked at 60 °C for 60 min. Sections were deparaffinized in xylene, dehydrated in ethanol, and air-dried. RNAscope^®^ 2.0HD Assay-Brown kits (ACD, Newark, CA, USA) were used, according to manufacturer’s instructions, to detect mRNA transcripts of CDA and hENT1: TMAs were heated to 100–104 °C in a citrate buffer to unmask target mRNA and permeabilize cells, followed by treatment with a protease inhibitor. The 15 probes for CDA hybridized between position 31 and 957 of the mature mRNA (NM_001785.2) and the 20 probes for hENT1 were designed to hybridize to the mature mRNA for gene SLC29A1 (NM_001078177.1) between positions 479 and 1774. PPIB and DapB were used as positive and negative controls. Probes were hybridized for 2 h at 40 °C. Signal amplification from the hybridized probes allowed detection of transcripts by 3,3′-diaminobenzidine. Counterstaining with hematoxylin localized the brown punctate dots within the cells.

### 2.4. Quantification Using RNAscope SpotStudio^®^ Software

Sections were scanned with an Aperio ScanScope^®^ microscopy scanner (Leica Microsystems [UK] Ltd., Milton Keynes, UK) at ×40 magnification. All cores were manually reviewed by FC, identifying cancerous regions. Damaged tissue was omitted, as were areas with debris/artifacts obscuring the area of interest. The pathologist and scientists involved were blinded to patient data, including treatment and outcome. RNAscope Spot Studio^®^ v 1.0 Software (ACD) was used to detect and count dots on a cell by cell basis over the entire cancerous region. Full details of the assessment can be found in the Appendix A. Parameters were the mean number of spots per cell (spot clusters were taken as equivalent to 10 spots when included) or proportion of cells with a given range of spots. The ranges were: Group 1, zero spots per cell, Group 2, 1–5 spots per cell and Group 3 ≥ 6 spots per cell. For each patient, the scores for individual cores were averaged as a mean value. The positive control, PPIB, showed detectable spots corresponding to individual transcripts whereas the negative control, DapB, had none. Following the computerized analysis of each TMA, a final manual quality check of every individual core was performed.

### 2.5. Immunohistochemistry

Immunohistochemistry with hENT1 antibody (10D7G2) or CDA antibody (ab137605, Abcam, Cambridge, UK) was performed as previously reported [23] The intensity of hENT1 cytoplasmic and membrane staining was scored by FC, accompanied by a research assistant (Elizabeth Garner), and H-scores were derived for each core ([intensity score] × [percentage of stained tumor cells]) with mean H-score calculated for each patient (Appendix A). For CDA an automated scoring system was used, described in Appendix A.

### 2.6. Statistical Analysis

Overall survival, measured from the date of randomization, was estimated using the method of Kaplan–Meier [38] with unadjusted differences between groups assessed using the log-rank test. Analyses were carried out using Cox proportional hazards models to assess the impact of biomarkers, individually and in combination, on overall survival. All models included tumor stage, lymph node involvement, and resection margins as prognostic factors with the effect of biomarkers nested within treatment effect. This allowed for the effects of prognostic clinical factors to be calculated across the patient cohort, whereas the effects of biomarkers are allowed to differ between treatment arms.

The assumption of proportionality was assessed via inspection of the Schoenfeld residuals. Comparing the residuals against the rank sum of time produced a global test for proportionality.

All statistical tests were two-sided and *p* < 0.05 was considered significant. All analyses were carried out using R version 3.3 (R Core Team).

## 3. Results

Tissue samples representing 290 out of 434 patients (67%) were of sufficient quality to allow scoring of mRNA. Restricting to patients that had matched protein hENT1 H-scores [23] gave 277 patients for the final analysis.

### 3.1. Determining Expression Levels of hENT1 and CDA in PDAC

CDA and hENT1 mRNA expression was detected only in epithelial cells and not in the surrounding stromal matrix. Representative images of RNA analysis are shown in Figure 1. Different assessment methods were compared and found to give equivalent ranking (high to low expression) for the patients (Appendix A). Mean single spots per cell excluding clusters (MSPC) were chosen for all analyses. The relationship between mRNA and protein expression was investigated in patientC cores where matched mRNA and protein data were available, this showed no correlation for either CDA or hENT1 (see Appendix A).

### 3.2. Univariate and Multivariable Analyses of Clinical and Pathological Characteristics

The ESPAC-3(v2) clinical trial was designed to show differences in survival according to treatment and so variation in clinicopathological features was minimized at trial randomization. However, univariate analysis by Cox proportional hazard regression, subdividing the chemotherapy treatment groups, showed that resection margin status (HR 1.56: 95% CI 1.20–2.03 *p* = 0.001), lymph node involvement (HR 1.94: 95% CI 1.39–2.71 *p* < 0.001) and tumor stage (HR 1.51: 95% CI 1.15–2.00 *p* = 0.004) were all significant prognostic factors for patients treated with 5-FU. Tumor diameter was a significant prognostic factor for gemcitabine (HR 1.64: 95% CI 1.14–2.39 *p* = 0.010), but did not reach statistical significance as a prognostic factor for 5-FU treated patients (HR 1.31: 95% CI 0.89–1.96 *p* = 0.177) (Table 1).

### 3.3. Overall Survival Analysis

CDA protein expression level was not found to be prognostic for either treatment group (Appendix A). hENT1 mRNA was not prognostic in patients treated with 5-FU; surprisingly there was a trend toward better survival in patients with low hENT1 mRNA treated with gemcitabine (Appendix A).

High expression of CDA mRNA conferred a poorer patient outcome regardless of chemotherapy; this was more pronounced in patients treated with 5-FU than in patients treated with gemcitabine. The median survival for patients treated with 5-FU expressing the upper tertile (high) CDA mRNA was 14.6 (95% CI = 8.4–24.1) months compared with 26.4 (95% CI = 21.4–29.7) months for the remaining patients, defined as low CDA expressers (χ^2^ = 5.18, *p* = 0.02) (Figure 2A). For patients treated with gemcitabine, high CDA expressers had a median survival of 21.2 (95% CI = 15.7–26.2) months compared with 24.8 (95% CI = 18.3–33.0) months for low CDA expressers (Figure 2B, χ^2^ = 5.14, *p* = 0.02).

hENT1 protein expression, as expected from previously published data [23], was prognostic for patients treated with gemcitabine, despite 20% fewer patients being included in the current analysis (HR = 0.60 (95% CI = 0.42–0.86), Wald χ^2^ = 7.90, *p* = 0.05). The median survival was 26.0 (95% CI = 21.2–32.8) months for high expressers (as defined previously [23]) compared with 16.8 (95% CI = 14.1–24.8) months for low expressers (χ^2^ = 7.58, *p* = 0.006) (Figure 2C,D). There was no correlation between high or low hENT1 protein expression and survival in 5-FU treated patients.

To rule out confounding factors and investigate the interaction between the biomarker combination and treatment, CDA mRNA expression was first considered as a continuous variable. With Cox regression this was shown to be significantly prognostic with 5-FU treatment (Table 2), giving an HR of 4.35 (95% CI = 1.14–16.62, *p* = 0.03). The same trend was seen in the gemcitabine arm but in this case, it did not reach statistical significance (HR=3.15, 95% CI = 0.93–10.68, *p* = 0.07). When the Cox model was used with CDA mRNA expression subdivided into upper tertile and the rest, the same trends were seen in both arms, but in this case, it reached significance for gemcitabine and not 5-FU (5-FU: HR 1.41 (95% CI = 0.91−2.17), *p* = 0.12; gemcitabine: HR 1.62 (95% CI = 1.12−2.39), *p* = 0.011). The lack of significance for the 5-FU arm was largely due to the impact of nodal status on the model.

Of the clinicopathological factors, only tumor stage correlated with CDA (Appendix A), although this association was weak. All models included tumor stage, lymph node involvement, and resection margin. A test of proportionality was carried out (see Statistical Methods), this was not significant [$\chi_{6}^{2}$ = 18.84, *p* = 0.096]. The term which had the biggest contribution towards non-proportionality was tumor stage. Removing this term had no effect on the model interpretation. Therefore, the tumor stage did not explain the relationship between CDA and survival (Table 2).

When CDA mRNA and protein were combined, the mRNA expression level, as expected, was prognostic in the 5-FU arm. Stratification with protein level made little difference. However, no significant prognostic effect of mRNA or protein was observed in the gemcitabine arm when the mRNA level was stratified by protein (Appendix A). The small numbers in the subgroups (dividing by treatment, mRNA, protein, and nodal status) meant that regression analysis was inappropriate. Stratification of hENT1 protein with hENT1 mRNA showed that the protein remained a prognostic marker in the gemcitabine arm, but this was only statistically significant where hENT1 mRNA was low (Appendix A).

From Table 2 we know that hENT1 is a predictive marker in a model incorporating CDA mRNA. The question remained whether a combination of CDA mRNA and hENT1 protein would give greater predictive power than hENT1 alone. All combinations are shown in Table 3. Kaplan-Meier survival curves confirm that the combined biomarkers were only significantly prognostic in the gemcitabine treatment arm (Figure 3A,B). Correspondingly, 5-FU gave a survival advantage over gemcitabine in patients with low hENT1 and low CDA (Figure 3C) while, gemcitabine gave a survival advantage where hENT1 is high and there is low CDA (Figure 3E), but where CDA is high, hENT1 seems to have negligible predictive value (Figure 3D,F). With gemcitabine, patients expressing low CDA mRNA with high hENT1 protein have the longest overall median survival of 28.0 (95% CI = 21.1–45.5) months compared with 23.8 (95% CI = 16.6−28.7) months in patients with high hENT1 protein and high CDA mRNA. When treated with 5-FU, patients with high hENT1 protein and low CDA mRNA have a median survival of 22.6 (95% CI = 16.9–29.6) months, and patients with high hENT1 protein and high CDA mRNA 20.1 months (95% CI = 5.0−37.5). In contrast, individuals with low CDA mRNA and low hENT1 protein do better with 5-FU: median survival 29.3 (95% CI = 21.9–41.9) months compared to survival of 18.3 (95% CI = 13.9–28.3) months with gemcitabine (Table 3). This confirms the previous reports that patients with low hENT1 would benefit from 5-FU rather than gemcitabine. However, patients with low hENT1 and high levels of CDA transcript have poor survival when treated with either gemcitabine or 5-FU (median 14.6 and 14.2 months respectively).

## 4. Discussion

Analysis of CDA mRNA expression showed that it was prognostic for both 5-FU and gemcitabine, with high expression of CDA mRNA correlating with poor survival, regardless of the type of chemotherapy. This was not seen with CDA protein; no discernible difference in survival between low and high CDA in either treatment group.

CDA protein may come from a variety of sources, including bacteria, furthermore secreted CDA could be lost from the extracellular space during tissue processing. The mRNA sequence assayed is specific for the product of the cancer cell’s CDA gene. As ESPAC-3(v2) was an adjuvant study, the survival of the patient will depend on the response of metastatic or residual cancer cells to therapy. These residual cells will reside in a different environment to the primary tumor, but inherent factors (e.g., genetic or epigenetic) that influence expression may be shared with the resected tumor cells. Indeed there is compelling evidence that driver mutations are generally maintained in metastases and heterogeneity is due mainly to the gain (or loss) of passenger mutations [39].

High levels of hENT1 protein are significantly associated with survival only in gemcitabine-treated patients [23]. hENT1 mRNA expression was not prognostic with 5-FU and for gemcitabine, the trend was for longer survival with low (not high) mRNA.

hENT1 mRNA and protein expression showed no correlation, as also reported in previous studies [40,41]. Indeed, Tavano et al. described an inverse relationship between protein and mRNA expression [42]. This suggests that post-transcriptional mechanisms determine protein levels in pancreatic tumor cells. The immunohistochemistry protocol with the 10D7G2 antibody provides the most informative prognostic information

High expression of CDA protein has been linked to gemcitabine resistance [35,43]. In our study CDA mRNA was associated with a worse prognosis with gemcitabine treatment but was also prognostic with 5-FU treatment. CDA may influence the flux of 5-FU metabolism and its toxicity. Salvage pathways involving orotate phosphoribosyl transferase (OPRT) play an important role in pancreatic cancer cell metabolism [44]; conversion of cytidine to uracil (and then orotate) by CDA will change the rate of salvage and the rate of 5-FU metabolism and uptake [45]. Alternatively, low CDA may associate with better outcomes for reasons completely independent of benefits from 5-FU or gemcitabine, for example, because there is a lower proliferation rate and therefore less nucleoside turnover in less aggressive tumors.

CDA mRNA levels made the greatest difference to survival in patients treated with 5-FU who had low hENT1. By contrast, for patients with low hENT1 treated with gemcitabine the impact of CDA was marginal. Perhaps gemcitabine concentration is so low in these cancer cells that no survival benefit for the patient is offered, hence there is no benefit to be lost by the action of CDA. Alternatively, a low level of hENT1 could result in deaminated gemcitabine not being rapidly transported out of cells, reducing the benefit of having low levels of the deaminase.

Empirically, patients with low hENT1 and low CDA survive significantly better if given 5-FU than if given gemcitabine: median overall survival with 5-FU 29.3 months (95% CI: 21.9–41.9) compared to just 18.3 months (95% CI:13.9–28.3). While, patients with high hENT1 benefit from gemcitabine over 5-FU, irrespective of CDA mRNA levels, it is clear that patients with low hENT1 and low CDA would benefit more from 5-FU. Patients with low hENT1 and high CDA appear not to benefit from gemcitabine or 5-FU with a median survival of just 14 months in either case. The recommendation for the selection of adjuvant therapy would be to first test for the hENT1 protein level. It can be assumed that patients with a high level of hENT1 would benefit from gemcitabine, while those with a low level should have an additional analysis for CDA transcript. A low level of CDA mRNA would support the use of 5-FU based therapy. One caveat to this recommendation is that at present only one antibody (10D7G2) is appropriate for measuring hENT1 level for this purpose [46] and this is in short supply. Development of both hENT1 IHC and CDA ISH is ongoing.

In this paper, we are considering a subset of the patients in the JAMA paper describing the full set of patients on the ESPAC 3(v2) clinical trial [5]. Indeed the requirement for data with both CDA and hENT1 means that this group of patients is even more restricted than the patients assessed in the original paper describing the predictive value for hENT1 [23]. Bias in the selection of the patients is a concern. Notable differences in comparison to the previous reports are that performance status, tumor grade, local invasion, and smoking all failed to reach significance in the current publication, all of these can be explained by a reduction in power due to smaller numbers.

A similar reduction in power was seen in our JNCI paper which identified hENT1 as a biomarker. Median survivals for patients treated with gemcitabine having low hENT1/high hENT1 protein in our original paper was 17.1/26.2 months. These values are very close to the observation of 16.8/26.0 months seen with our more restricted population. For 5FU treated patients the values of 25.6/21.9 months in the previous paper were a little further from the values seen here (22.6/24.1 months), but still not suggestive of any particular bias.

## 5. Conclusions

We have demonstrated that patients stratified for adjuvant treatment with gemcitabine using hENT1 protein can be further stratified using CDA transcript level. The benefit of 5-FU over gemcitabine in patients with low hENT1 protein is lost in patients with high CDA. Further work is required to see how this can be applied to patients treated with combination therapies such as FOLFIRINOX, gemcitabine with nab-paclitaxel, or gemcitabine with capecitabine.

## Figures and Tables

**Figure 1 cancers-13-05758-f001:**
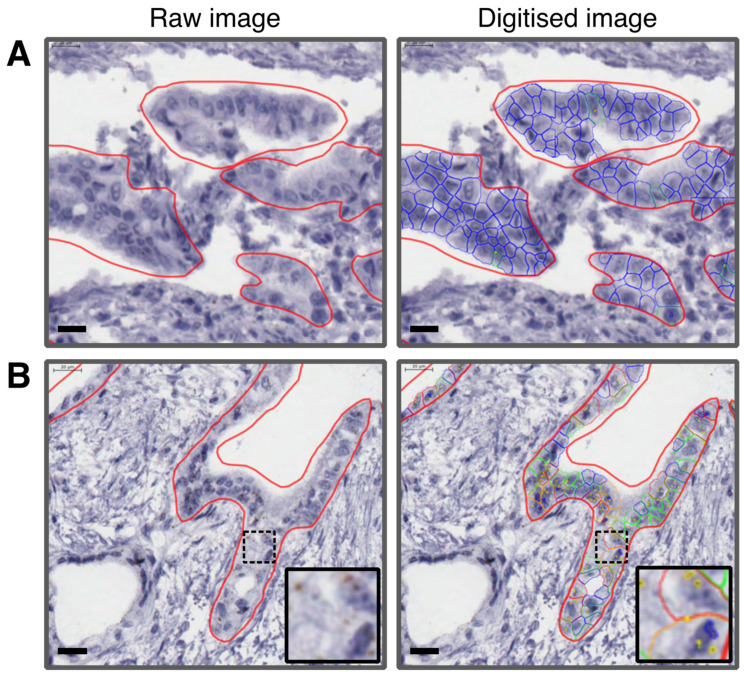
Representative images of tissue cores showing CDA mRNA expression before and after *SpotStudio^®^* analysis. (**A**) Raw and digitized images of tissue expressing low CDA mRNA. (**B**) Raw and digitized images of tissue expressing high CDA, with inset example of cells with high CDA expression. Thick red line surrounds areas with tumor cells. Blue lines = Cells without any spots. Green lines = Cells with one single spot. Orange lines = Cells with between two and five spots. Thin red lines = Cells with six or more spots. Scale bar = 20 µm.

**Figure 2 cancers-13-05758-f002:**
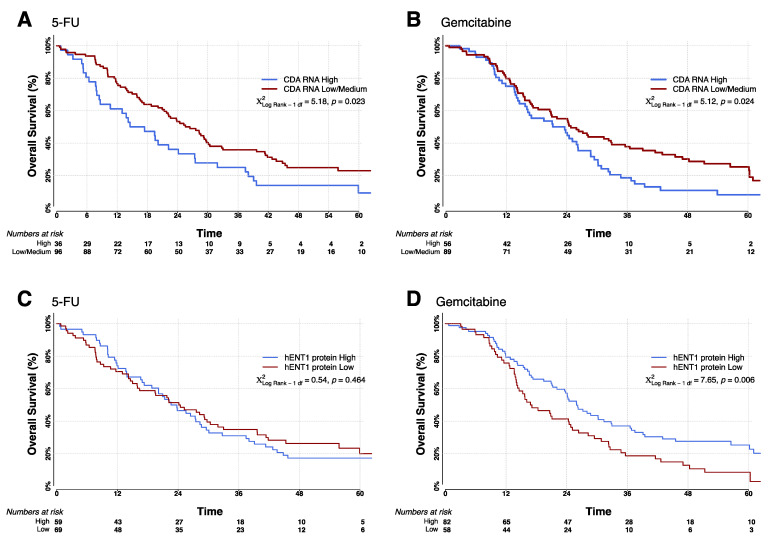
Kaplan-Meier survival curves separated by both treatment arms (5-FU and gemcitabine) and biomarker expression levels. (**A**,**B**): Survival curves for patients with low and high CDA mRNA levels (low ≤ 0.61, high > 0.61 MSPC), for patients randomized to adjuvant treatment with 5-FU (**A**) and gemcitabine (**B**). (**C**,**D**): Survival curves for low and high hENT1 expressing patients (low ≤48, high >48 H-Score), for patients randomized to adjuvant treatment with 5-FU (**C**) and gemcitabine (**D**). All groups and the number of at-risk individuals are shown in each graph. All *p*-values were determined by log-rank analyses using two-sided χ^2^ tests.

**Figure 3 cancers-13-05758-f003:**
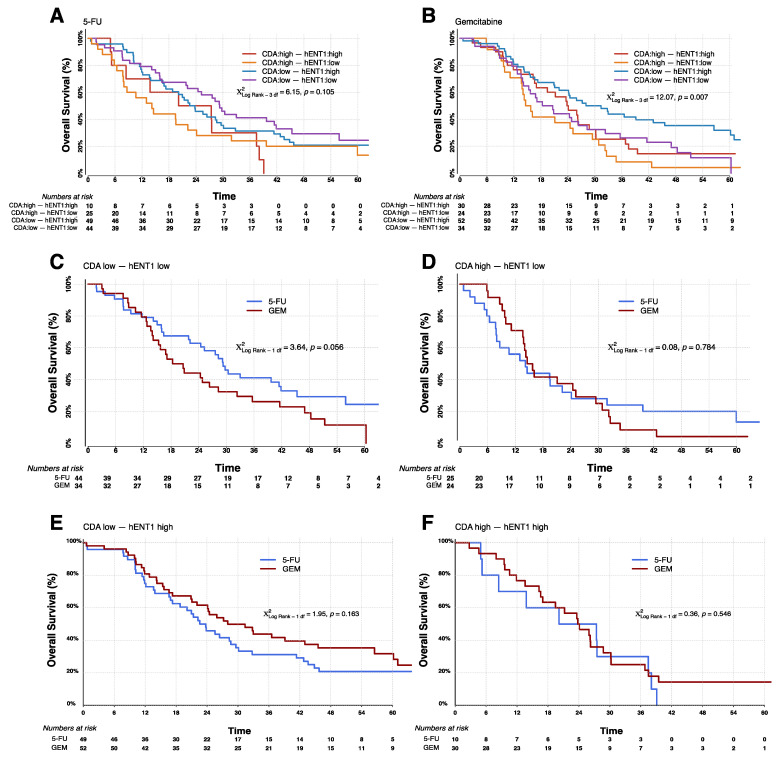
Kaplan-Meier survival curves for analyses of combined CDA mRNA and hENT1 protein biomarker interaction looking at all expression level combinations (CDA low, hENT1 low; CDA high, hENT1 low; CDA low, hENT1 high; CDA high, hENT1 high). Graphs show response to treatment with 5-FU (**A**) and gemcitabine (**B**). In (**C**,**D**) the same data as above is presented showing the difference in survival for patients treated with 5-FU compared to gemcitabine in patients with low hENT1 protein and either low CDA (**C**) or high CDA (**D**), and patients with high hENT1 protein and either low CDA (**E**) or high CDA (**F**). All groups and the number of at-risk individuals are shown for each graph. All *p*-values were determined by log-rank analyses using two-sided χ^2^ tests.

**Table 1 cancers-13-05758-t001:** Univariate analysis of clinical and pathological factors in the 5-FU and gemcitabine arms.

		Summary Statistics		Hazard Ratio (95% Confidence Interval)		
Characteristic	Level	5-Fluorouracil	Gemcitabine	5-Fluorouracil	Gemcitabine	Total
**Resection Margin**				*n* = 132	*n* = 145	*n* = 277
	Negative	70 (53%)	86 (59%)	1	1	1
	Positive	62 (47%)	59 (41%)	1.80 (1.23–2.63)	1.35 (0.94–1.94)	1.56 (1.20–2.03)
				Wald χ^2^ = 9.05, ***p =* 0.003** *	Wald χ^2^ = 2.62 *p =* 0.106	Wald χ^2^ = 11.22, ***p =* 0.001**
**WHO**				*n* = 132	*n* = 145	*n* = 277
	0	52 (39%)	49 (34%)	1	1	1
	1	67 (51%)	80 (55%)	1.26 (0.86–1.96)	1.40 (0.94–2.07)	1.33 (1.01–1.76)
	2	13 (10%)	16 (11%)	0.68 (0.29–1.59)	1.24 (0.68–2.25)	0.92 (0.56–1.53)
				Wald χ^2^ = 2.89, *p =* 0.236	Wald χ^2^ = 2.75 *p =* 0.253	Wald χ^2^ = 5.24, *p =* 0.073
**Lymph Node Status**				*n* = 132	*n* = 145	*n* = 277
	Negative	29 (22%)	29 (20%)	1	1	1
	Positive	103 (78%)	116 (80%)	2.40 (1.47–3.90)	1.56 (0.99–2.46)	1.94 (1.39–2.71)
				Wald χ^2^ = 12.30, ***p =* 0.001**	Wald χ^2^ = 3.63, *p =* 0.057	Wald χ^2^ = 15.18 ***p* < 0.001**
**Tumor Stage**				*n* = 131	*n* = 144	*n* = 275
	01/02	38 (29%)	46 (32%)	1	1	1
	03/04	93 (70%)	98 (68%)	1.69 (1.10–2.59)	1.39 (0.95–2.01)	1.51 (1.15–2.00)
				Wald χ^2^= 5.81, ***p =* 0.016**	Wald χ^2^ = 2.95, *p =* 0.086	Wald χ^2^ = 8.47, ***p =* 0.004**
**Tumor Grade**				*N* = 129	*n* = 142	*n* = 271
	Well	7 (5%)	10 (7%)	1	1	1
	Moderately	89 (67%)	98 (68%)	0.60 (0.37–0.96)	0.83 (0.44–1.58)	0.72 (0.47–1.01)
	Poorly	33 (25%)	34 (23%)	0.67 (0.35–1.28)	1.19 (0.58–2.44)	0.91 (0.55–1.51)
				Wald χ^2^ = 4.54, *p =* 0.103	Wald χ^2^ = 2.75, *p =* 0.753	Wald χ^2^ = 3.60 *p =* 0.165
**Local Invasion**				*n* = 132	*n* = 142	*n* = 274
	No	73 (55%)	72 (50%)	1	1	1
	Yes	59 (45%)	70 (48%)	1.25 (0.85–1.84)	1.10 (0.77–1.58)	1.17 (0.90–1.52)
				Wald χ^2^ = 1.31, *p =* 0.252	Wald χ^2^ = 0.26, *p =* 0.607	Wald χ^2^ = 1.37 *p* = 0.242
**Maximum Tumor diameter**				*n* = 128	*n* = 139	*n* = 267
	<30 mm	69 (52%)	58 (40%)	1	1	1
	≥30 mm	59 (45%)	81 (56%)	1.31 (0.89–1.95)	1.64 (1.13–2.39)	1.47 (1.12–1.92)
				Wald χ^2^ = 1.82, *p =* 0.177	Wald χ^2^ = 6.73, ***p =* 0.010**	Wald χ^2^ = 7.84 ***p =* 0.005**
**Diabetes mellitus**				*n* = 129	*n* = 141	*n* = 270
	No	102 (77%)	106 (73%)	1	1	1
	Yes	27 (20%)	35 (24%)	1.06 (0.65–1.75)	0.99 (0.64–1.53)	1.02 (0.74–1.41)
				Wald χ^2^ = 0.07, *p =* 0.797	Wald χ^2^ = 0.00, *p =* 0.951	Wald χ^2^ = 0.01, *p =* 0.905
**Gender**				*n* = 132	*n* = 145	*n* = 277
	Male	75 (57%)	88 (61%)	1	1	1
	Female	57 (43%)	57 (39%)	0.88 (0.59–1.32)	1.25 (0.86–1.81)	1.06 (0.81–1.39)
				Wald χ^2^= 0.38, *p =* 0.537	Wald χ^2^ = 1.42, *p =* 0.234	Wald χ^2^ = 0.16, *p =* 0.686
**Age, years**				*n* = 132	*n* = 145	*n* = 277
	≥64	65 (49%)	80 (55%)	1	1	1
	<64	67 (51%)	65 (45%)	1.37 (0.93–2.02)	0.84 (0.58–1.21)	1.07 (0.83–1.40)
				Wald χ^2^= 2.55, *p =* 0.110	Wald χ^2^ = 0.90, *p =* 0.342	Wald χ^2^ = 0.28, *p =* 0.598
**Smoking**				*n* = 125	*n* = 128	*n* = 253
	Never	52 (39%)	58 (40%)	1	1	1
	Ex	51 (39%)	52 (36%)	0.93 (0.60–1.43)	1.08 (0.70–1.66)	1.00 (0.74–1.35)
	Current	22 (17%)	18 (12%)	0.91 (0.53–1.57)	1.76 (1.02–3.05)	1.20 (0.81–1.78)
				Wald χ^2^ = 0.17, *p =* 0.917	Wald χ^2^ = 4.16, *p =* 0.125	Wald χ^2^ = 0.95, *p =* 0.623

* Significant values in bold.

**Table 2 cancers-13-05758-t002:** Cox regression analysis of biomarkers in the 5-FU and gemcitabine arms.

		5FU			Gemcitabine		
Characteristic		est (se)	HR (95% CI)	Pval	est (se)	HR (95% CI)	*p*-Value
**Single biomarker: CDA or hENT1 combined with stage, resection margin and lymph node involvement**							
CDA mRNA Expression	Per unit increase in MSPC	1.47 (0.685)	4.35 (1.14, 16.62)	0.032	1.15 (0.623)	3.15 (0.93, 10.68)	0.066
CDA mRNA Expression (Low vs. High)	Low		1 (Reference)			1 (Reference)	
	High	0.34 (0.221)	1.41 (0.91, 2.17)	0.120	0.49 (0.192)	1.62 (1.12, 2.39)	**0.011** *
hENT1 protein expression	Per unit increase in (log) H-score	−0.03 (0.159)	0.97 (0.71, 1.33)	0.861	−0.26 (0.129)	0.77 (0.60, 0.99)	**0.047**
hENT1 protein expression (Low vs. High)	Low		1 (Reference)			1 (Reference)	
	High	0.18 (0.207)	1.21 (0.80, 1.81)	0.374	−0.43 (0.192)	0.65 (0.45, 0.95)	**0.025**
**Multiple biomarker: CDA and hENT1 combined with stage, resection margin and lymph node involvement**							
CDA mRNA Expression (Low vs. High)	Low		1 (Reference)			1 (Reference)	
	High	0.39 (0.226)	1.48 (0.95, 2.31)	0.082	0.50 (0.193)	1.65 (1.13, 2.41)	**0.009**
hENT1 protein expression (Low vs. High)	Low		1 (Reference)			1 (Reference)	
	High	0.28 (0.212)	1.32 (0.87, 1.99)	0.193	−0.41 (0.191)	0.66 (0.46, 0.96)	**0.030**

* Significant values in bold.

**Table 3 cancers-13-05758-t003:** Median overall survival in subgroups split by treatment arm, 5-FU or gemcitabine (GEM), cytidine deaminase (CDA) mRNA, and human equilibrative nucleotide transporter-1 (hENT1) protein status.

Arm	Biomarker Expression (High or Low)	Number	Median OS	95% Confidence Interval	Log Rank	*p*-Value
5-FU/FA	**CDA High**	36	14.6	8.4–24.1	5.17	**0.0229**
	**CDA Low**	96	26.4	21.4–29.7		
GEM	**CDA High**	56	21.2	15.7–26.2	5.14	**0.0234**
	**CDA Low**	89	24.8	18.3–33.0		
5-FU/FA	**hENT1 High**	59	22.6	17.3–28.6	0.53	0.4658
	**hENT1 Low**	69	24.1	15.9–30.4		
GEM	**hENT1 High**	82	26.0	21.2–32.8	7.58	**0.0059**
	**hENT1 Low**	58	16.8	14.1–24.8		
5-FU	**CDA Low, hENT1 Low**	44	29.3	21.9–41.9	6.14	0.1050
	**CDA High, hENT1 Low**	25	14.2	7.9–24.1		
	**CDA Low, hENT1 High**	49	22.6	16.9–29.6		
	**CDA High, hENT1 High**	10	20.1	5.0–37.5		
GEM	**CDA Low, hENT1 Low**	34	18.3	13.9–28.3	12.0	**0.0073**
	**CDA High, hENT1 Low**	24	14.6	11.1–25.1		
	**CDA Low, hENT1 High**	52	28.0	21.1–45.5		
	**CDA High, hENT1 High**	30	23.8	16.6–28.7		

Significant values in bold.

## Data Availability

The data presented in this study are available on request from the corresponding author. The data are not publicly available due to blinding of authors to trial data when carrying out experimental analysis.

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
