# Peer review of "hENT1 Predicts Benefit from Gemcitabine in Pancreatic Cancer but Only with Low CDA mRNA"

_cancers, 2021, doi:10.3390/cancers13225758_

Round 1
Reviewer 1 Report
In this manuscript, Aughton and colleagues assessed the combined predictive value of cytidine deaminase (CDA) with human equilibrative nucleoside transporter 1 (hENT1) expression in patients treated with 5-fluorouracil or gemcitabine in the ESPAC-3(v2) trial population. This article provided an extension of analysis, results, and conclusions over previous studies. The group of research possesses enough seniority in the topic for writing this article. I only have several minor comments:
- Authors should follow the style of a structured abstract, which is based on the IMRAD structure of a paper, but without using headings. (https://www.mdpi.com/authors/layout#_bookmark5)
- Probes information (regions) for CDA and hENT1 RNA detection need to be listed in a table.
- Line 184-185: The addition of statistical information (HR, 95% CI, p-value) allows readers to better understand the raw data.
Author Response
Thank you for the very positive comments. We have removed the headings within the abstract, added hazard ratios, 95% confidence intervals and p-values for the statements in lines 184 to 186 and added the positions of the probes into the materials and methods between lines 120 and 123
Reviewer 2 Report
In this MS authors suggest that high hENT1 protein was prognostic with gemcitabine with increased median overall survival rate independent of low CDA mRNA by clinical trial.
Despite interesting data, it did not show any novelty compared to previous evidence., since this group published similar paper in Cancers in 2021.
William Greenhalf et al Pancreatic cancer hENT1 expression and survival from gemcitabine in patients from the ESPAC-3 trial, J Natl Cancer Inst, 2014 Jan;106(1):djt347
Author Response
We would like to thank the reviewer for their consideration of the manuscript. The other publications referred to were extensively referenced in the manuscript and we feel that the additional unpublished information is clearly identifiable in the revised manuscript and this new information is the basis of the conclusions and discussion presented.
Reviewer 3 Report
Aughton K et al. examined the relevance of two potential biomarkers among patients with pancreatic cancer receiving adjuvant gemcitabine or 5-FU. They carried out a study using tissue samples collected within the ESPAC-3(v2) randomized clinical trial and found that cytidine deaminase (CDA) expression combined with equilibrative nucleoside transporter 1 (hENT1) could predict response to chemotherapy. Generally, the paper is well planned and addresses a clinically relevant topic. Although similar studies related to CDA and hENT1 have already been published, the current results obtained from a RCT may provide additional insight into the role of both biomarkers.
Minor points:
- The authors should discuss about the differences in prognostic factors identified in the present study and in the original study published in JAMA (2010). These discrepancies may suggest some selection bias between these two patient populations.
- The authors should comment on their variable selection process for the Cox model. They used a combination of stage (i.e. incorporating both pT and pN categories), resection margin and lymph node involvement (pN). Therefore, it is unclear why they used stage and nodal status (risk of collinearity) instead of stage alone or the combination of pT and pN categories.
- Local invasion in Table 1 should be explained.
- The authors should provide details for CDA and hENT probes used for in situ hybridization
Author Response
We would like to thank the revewer for their comments. We agree that discussion of the differences seen in the current paper and previous papers need to be addressed. We have added the following to the discussion.
'In this paper we are considering a subset of the patients in the JAMA paper describing the full set of patients on the ESPAC 3(v2) clinical trial5. Indeed the requirement for data with both CDA and hENT1 means that this group of patients is even more restricted than the patients assessed in the original paper describing the predictive value for hENT123. Bias in the selection of the patients is a concern. Notable differences in comparison to the previous reports are that performance status, tumour grade, local invasion and smoking all failed to reach significance in the current publication, all of these can be explained by a reduction in power due to smaller numbers.
A similar reduction in power, was seen in our JNCI paper which identified hENT1 as a biomarker. Median survivals for patients treated with gemcitabine having low hENT1/high hENT1 protein in our original paper was 17.1/26.2 months. These values are very close to the observation of 16.8/26.0 months seen with our more restricted population. For 5FU treated patients the values of 25.6/21.9 months in the previous paper were a little further from the values seen here (22.6/24.1 months), but still not suggestive of any particular bias.'
The reviewer's comments with regard to the variable selection process for the Cox model is astute. The analytical approach taken was to construct a 'marker naive' Cox proportional hazards model which consisted of key demographic/prognostic covariates and to act as a basis upon which the impact of hENT1 and CDA could be evaluated (and not to evaluate clinical parameters and their impact on survival). This initial model was constructed using a backwards step-wise procedure based on Akaikes Information Criterion (AIC). This procedure selected both T stage and pN in the model. Upon observing this, we attempted a number of alternative models (e.g. removing pN, including tumour size in place of 'Stage') evaluating each model likelihood and parameter estimates. No alternative models provided a better overall fit and there was negligible impact on any coefficients associated with covariates included in the model (e.g. the coefficient associated with pN were similar irrespective of whether 'Stage' was included in the model). As the primary aim was to develop a 'basis model' which explained as much as the variability in the data as is possible prior to investigating the effects of CDA and hENT1, we opted to retain the model that was obtained from the step-wise procedure, as the best practical basis model.
Local invasion was considered as positive if there was any involvement of organs adjacent to the pancreas (typically the duodenum) or vasculature.
We have added details of the probes used to evaluate CDA and hENT1 to the methods section between lines 120 and 123 as below:
'The 15 probes for CDA hybridized between position 31 and 957 of the mature mRNA (NM_001785.2) and the 20 probes for hENT1 were designed to hybridize to the mature mRNA for gene SLC29A1 (NM_001078177.1) between positions 479 and 1774. PPIB and DapB were used as positive and negative controls.'
Round 2
Reviewer 2 Report
They addressed well to my comments.